# Learning from Forest Trees: Improving Urban Tree Biomass Functions

**Christian Vonderach [1,2,*,†] and Adrienne Akontz [3,†]**

[1] Forest Research Institute Baden-Württemberg, 79100 Freiburg, Germany
[2] Chair of Forest Growth and Dendroecology, University of Freiburg, Tennenbacherstraße 4, 79106 Freiburg, Germany
[3] TreeConsult Brudi & Partner, 82131 Gauting, Germany; a.akontz@tree-consult.org
[*] Correspondence: christian.vonderach@forst.bwl.de; Tel.: +49-761-4018-208
[†] These authors contributed equally to this work.

**Abstract:** Trees are one of the few carbon sinks in urban areas. Different methods are available to assess the biomass of urban trees, one of these being allometric biomass functions. Biomass functions are well investigated, reliable and easy to apply if the required information is available. Our goal is to use biomass functions to enhance urban forest management tools with information on stored biomass and carbon. In this study, we test several approaches to estimate new species-specific biomass functions. We include data from urban and traditional forest trees since both origins can be modeled by the allometric relationship solely giving different parameter estimates. The tested models include mixed allometric models for urban trees only, the adjustments of available forest tree biomass models and a cross-classified mixed model (CCMM) using both data from urban and forest trees. We then show by cross validation that the CCMM, statistically separating the data into different species and origins, shows greater improvement over the simpler models. Hence, we state that the inclusion of forest tree data improves the performance on biomass predictions for urban trees: the urban tree biomass functions "learned from forest trees". The CCMM is also compared against the predictions of the above-ground biomass functions applied in the German National Forest Inventory. Comparable RMSE and slightly lower BIAS values are found, both for deciduous and coniferous tree species. With the approach of a cross-classified model, we also enable predictions for non-observed conifers in urban space, assuming comparable growth differences between deciduous and conifer species in forest stands and urban areas. A sample application using the CCMM model shows results for a small subset of data of an urban tree inventory, collected in a residential area in the city of Munich, Germany. It is applied to estimate carbon storage at two points in time and, hence, carbon fluxes in the period under consideration. Such information can help in the decision making and management of urban trees.

**Keywords:** biomass functions; cross-classified mixed model; urban trees; allometric model; carbon sink

## 1. Introduction

Urban trees provide several ecosystem services, one of them being a carbon storage [1–3]. During the last decades, a lot of research has been undertaken to bring light to the role of urban forests for carbon sequestration (e.g., [4–8] and many others). In comparison to traditional forests, the tree density in urban forests is much lower, but nevertheless urban trees represent a relevant reservoir of carbon and eventually a sink of carbon dioxide in the urban context (e.g., [9,10]). While urban tree management aims at maintaining or even enhancing the more obvious urban tree functions (e.g., shading, aesthetics, recreation, see also Konijnendijk et al. [1]), it might additionally focus on increasing the amount of stored carbon when it comes to climate protection. This requires methods for determining the current carbon storage to be able to evaluate the development of this reservoir.



Knowledge about the urban carbon reservoir and possibly the associated fluxes helps in understanding the role of urban trees in carbon dioxide emission mitigation for single trees (e.g., [11]), on the local [8,10,12–18], or nation-wide scale [2,4,6,19,20], to deduce the total net balance for a specific area [9,21,22], to conduct demand–supply analysis [21,23,24] for reporting in the context of the United Nations Framework Convention for Climate Change (UNFCCC) [6], to assess their potential in carbon credit markets (e.g., [25]), to draw comparisons to other urban (e.g., [17]) or traditional forests (e.g., [19]), build storage maps [13,26] or to develop management options [26,27] with regard to climate change and carbon dioxide mitigation.

For the estimation of stored carbon in urban space, several methods were already developed and applied. On the one hand, there are approaches using remote sensing [6,8,10,12,13,15,17], either based on airborne photogrammetry and/or airborne lidar. Ground-based lidar, i.e., terrestrial laser scanning (TLS), was applied to estimate single trees biomass (e.g., [5,7,23,28,29]). On the other hand, the well-established method of allometric biomass functions is used to estimate tree biomass, which might be upscaled to the desired spatial level. In some cases, predictors were deduced from remote sensing analysis [10,13,15,17]. Other studies developed specific biomass functions (e.g., [30,31] for small trees and Korea, respectively), used equations from nearby locations (e.g., [32]), or used forest tree biomass function for comparison (e.g., [16]). One obvious use case for biomass functions is to estimate biomass based on data and attributes stored in a tree inventory, if respective predictors and functions are available. For such a case, we want to develop biomass functions and apply these models to already available data. With that, we can enhance existing management tools by easy means. Remote sensing information is not easily available and accessible in a permanent manner for such management systems, yet.

Since the amount of stored biomass and, hence, carbon (being approximately half of the biomass) cannot be easily determined using existing forest tree biomass functions (e.g., [29,33] and also shown by this study), there is a need to develop specific functions for urban trees. Likewise, McPherson et al. [34] (p. 8ff) even point out that several factors might influence tree growth within and between cities. In this context, the authors highlight management practice, which also depend on regulatory rules. They also state that urban trees exhibit higher variability in habitus than rural trees. Russo et al. [14] and Strohbach and Haase [13] also advise to check applicability and comparability, respectively, when existing methods are applied to new areas. Hence, if new biomass functions for urban trees are to be developed, this guidance should be kept in mind and a trade-off made between generality and specificity of the situation at hand.

Biomass functions have already been used for a long time, and an extensive body of literature exists (see, e.g., [35–37]). Usually, biomass functions relate easy-to-measure variables, such as diameter in breast height (dbh) and tree height to total or component mass. Due to this relation, such biomass functions are also termed allometric biomass functions (see also [38]). In urban areas, one of the main difficulties in developing urban tree biomass functions is the scarcity of actual measured biomass data since cutting urban trees is usually not desired. In this study, we aim to develop a set of urban tree biomass functions to estimate the total above-ground biomass (agb) to be able to calculate stored carbon in urban tree stands using single tree attributes from well-established management tools, like communal tree inventories. Additionally, we show a modeling approach capable of improving these functions by including not only urban tree data but also forest tree data. In the following, we describe the two data sets for model building and present the applied regression models. Several alternative models are tested, and the leave-one-out cross-validation results favor the cross-classified mixed model. We discuss the different approaches and apply the proposed cross-classified mixed model on example data from a tree inventory in Munich.

## 2. Materials and Methods

### 2.1. Data

In this study, we could revert on previously generated biomass data from both urban trees [10] and forest trees [39]. Here, the term "urban trees" refers to rather free-standing trees grown in urban areas, like streets, gardens and parks. Excluded are trees from small stand-like structures occurring in urban areas and trees in rural areas. In contrast, "forest trees" refer to trees grown under light competition in semi-natural stands of larger size outside the city border and are typically subject to traditional forest management, like German productive forest stands. In the same sense, we use the terms "urban forests" and "traditional forests" throughout the text. The urban tree data set contains 164 non-destructively sampled urban trees of fourteen different tree species from a study conducted between 2009 and 2011 in Karlsruhe, Germany. These trees were measured by skilled arborists using the randomized branch-sampling (RBS) protocol (c.f. [40–42]). The study of Kändler et al. [10] sampled trees for all type of sites, including streets, gardens and parks based on a pre-selection by orthophotos and cadastre information. This assured the presence of weak, medium and strong trees, the representativeness of the tree species according to the tree register and the selection of trees from all parts of the city. Sampling took place during the end of January and the beginning of March 2010.

Using RBS, the main bole was measured at 0.5 m and 1 m above ground, followed by 2 m-sections up to the crown. Inside the tree crown, each of three paths from crown base to bud end were measured for segment length as well as bottom and top diameter between all knots, i.e., branching points. The total volume was then estimated by expanding measured segment volume by the path-wise cumulated selection probability based on all branch base diameters at each knot (for details, see [40,42]). Aggregated volume is transferred to biomass using specific gravity values of the respective tree species from literature [43]. The collected data include information about total above-ground biomass, several diameters along the stem (of which, here we use the diameter in 1 m height above ground, further called *d1*), tree height (*h*), height of green crown (*hgc*) and remotely sensed crown diameter (*cd*, see [10]). The required *d1* diameter was not available for all trees due to early branching. Hence, only 144 complete observations were available, spread over fourteen deciduous species, each holding three to thirty-two observations. Further biometric information is given in Table 1, and a graphical overview is given in Figure 1.

**Table 1.** Overview on sampled urban trees per species. Given are the number of sampled trees per species (n), mean values for diameter in breast height (*dbh*, measured in 1.3 m above ground, cm), diameter in 1 m above ground (*d1*, cm), tree height (*h*, m), height of green crown (*hgc*, m), crown diameter (*cd*, m) and above-ground biomass (*agb*, kg), calculated from volume and specific gravity.

| Species | Name | n | dbh | d1 | h | hgc | cd | agb |
|---|---|---|---|---|---|---|---|---|
| | | [n] | [cm] | [cm] | [m] | [m] | [m] | [kg] |
| *Acer campestre* | field maple | 5 | 21.6 | 21.9 | 8.8 | 2.2 | 6.1 | 206.3 |
| *Acer platanoides* | Norway maple | 32 | 47.6 | 48.4 | 13.9 | 2.3 | 10.5 | 1372.5 |
| *Acer pseudoplatanus* | sycamore maple | 3 | 47.0 | 48.0 | 15.6 | 2.5 | 10.6 | 1267.2 |
| *Aesculus hippocastanum* | sweet chestnut | 6 | 53.6 | 52.8 | 12.7 | 2.7 | 10.4 | 1658.9 |
| *Betula spp.* | birch spp. | 3 | 40.9 | 43.0 | 17.3 | 3.9 | 8.2 | 866.8 |
| *Carpinus betulus* | common hornbeam | 12 | 30.0 | 31.3 | 11.5 | 2.6 | 7.1 | 570.5 |
| *Fraxinus excelsior* | common ash | 15 | 48.9 | 50.3 | 14.7 | 3.0 | 10.6 | 2110.6 |
| *Platanus × acerifolia* | London plane | 14 | 79.2 | 83.4 | 20.8 | 3.6 | 12.1 | 4601.7 |
| *Prunus avium* | wild cherry | 4 | 25.4 | 26.0 | 12.2 | 2.9 | 4.6 | 308.6 |
| *Quercus robur* | common oak | 21 | 40.5 | 42.1 | 13.8 | 2.9 | 9.1 | 1750.4 |
| *Quercus rubra* | red oak | 8 | 75.6 | 75.8 | 19.6 | 2.5 | 15.6 | 5319.6 |
| *Robinia pseudoacacia* | black locust | 6 | 37.5 | 38.9 | 14.5 | 3.9 | 7.3 | 1406.9 |
| *Tilia × euchlora* | Caucasian lime | 7 | 49.9 | 50.9 | 14.9 | 3.5 | 8.4 | 1412.6 |
| *Tilia cordata* | small-leafed lime | 8 | 29.6 | 31.5 | 12.9 | 3.7 | 7.0 | 392.0 |

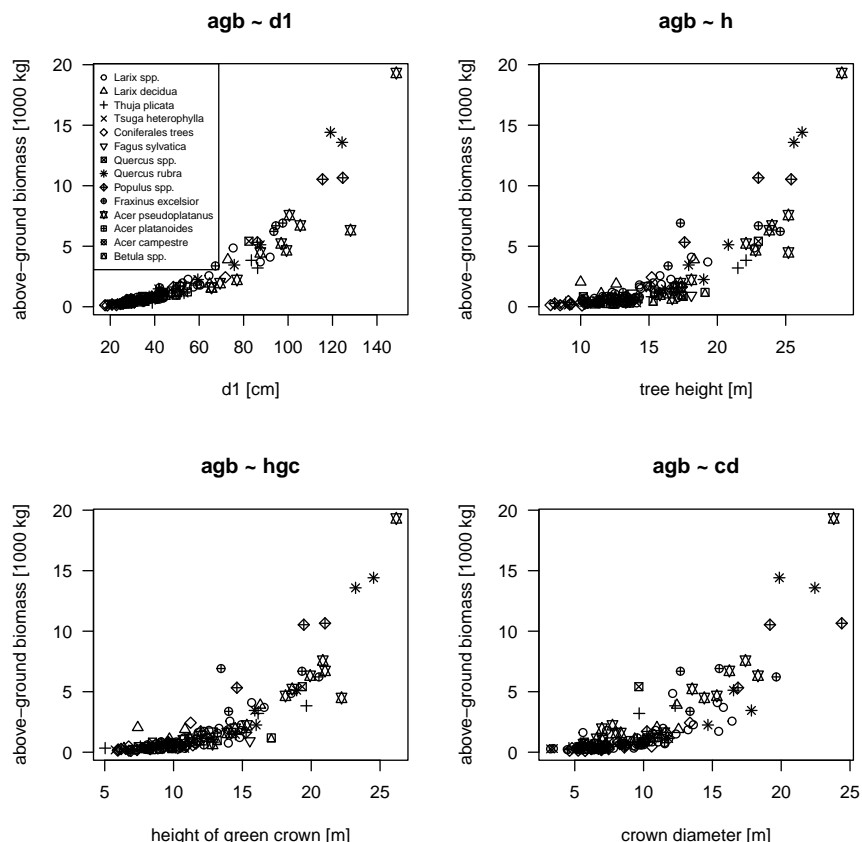

**Figure 1.** Relation between predictors and above-ground biomass (*agb*) for different species. Legend for all graphs is given in top left panel.

With regard to the auxiliary data of forest tree species, data from a German meta-study aiming at building additive component biomass functions for forests tree species were used [39]. These data (n = 2061 complete observations) originate from different sources (both with regard to geographic origin and sampling method), but a substantial part of the trees was also sampled using RBS (in contrast to the urban trees, forest trees were cut before sampling). The data contain the same variables as the urban tree data set, except *hgc* and remotely sensed *cd*. The missing *d1* was estimated based on the other predictors using the taper curve library BDAT [44,45]. Here, eight tree species, including four conifers, are given, holding 25 to 666 observations (for a data summary, see Table 2).

**Table 2.** Overview on sampled forest trees per species. Species names with asterisk (*) indicate availability in both data sets. The number of sampled trees per species, mean values for diameter in breast height (*dbh*, cm), diameter 1 m above ground (*d1*, cm), tree height (*h*, m) and above-ground biomass (*agb*, kg) are given. Data were compiled by [39] but originate from [46–50].

| Species | Name | n | dbh | d1 | h | agb |
|---|---|---|---|---|---|---|
| | | [n] | [cm] | [cm] | [m] | [kg] |
| *Abies alba* | silver fir | 29 | 41.8 | 43.4 | 25.8 | 1255.2 |
| *Acer pseudoplatanus* | sycamore maple * | 25 | 28.3 | 29.1 | 22.6 | 493.2 |
| *Fagus sylvatica* | European beech | 666 | 32.2 | 33.1 | 24.8 | 1078.8 |
| *Fraxinus excelsior* | common ash * | 37 | 33.2 | 34.2 | 25.6 | 1144.5 |
| *Picea abies* | Norway spruce | 616 | 32.7 | 34.3 | 24.9 | 581.0 |
| *Pinus sylvestris* | Scots pine | 311 | 31.7 | 32.9 | 23.0 | 516.2 |
| *Pseudotsuga menziesii* | Douglas fir | 130 | 31.0 | 32.0 | 24.7 | 580.5 |
| *Quercus* spp. | oak spp. * | 247 | 31.9 | 32.9 | 22.9 | 955.7 |

### 2.2. Methods

Aiming at stable biomass functions for at least the available fourteen tree species, we passed through several steps of model building. In short, we fitted allometric models for only the urban tree data, followed by a implementation of adjusted forest tree biomass functions to be able to predict stored biomass and carbon also for coniferous tree species. Finally, we implemented a common model for the urban and forest tree. All steps are detailed further in the following sections.

The urban tree data set consists of fourteen tree species (Table 1). We decided for a mixed model approach, because the number of observations is rather low for almost all species, which makes it difficult to receive stable biomass models for each species individually. Hence, we modeled the above-ground biomass (*agb*) for all species in one model, but allowed for structured deviations from this population average, given the factor "species" (*spp*). Additionally, we chose to fit the models on the (nonlinear) data scale to avoid the need of back-transformation and bias correction (c.f. [51]). We started by modeling the response *y* (= *agb*) by mixed allometric models of the general form

$$y = (\alpha + a) \prod_{i=1}^{p} X_i^{\beta_i + b_i} + \epsilon \tag{1}$$

with *p* predictors $X_i, i = 1 \ldots p$ and fixed-effect parameters $\alpha$ and $\beta_i$ for the population average, random terms $a$ and $b_i$ for species-specific deviations and investigated *d1* and further predictors, namely tree height (*h*), height of green crown (*hgc*) and crown diameter (*cd*). The known phenomenon of heteroscedastic errors in biomass data was treated by modeling the increasing variance species-wise as power of a variance covariate $v$, using *d1* or the estimated *agb* ($\hat{y}$) (c.f. Table 3, column 4, defining $v$), i.e., $var(\epsilon) = \sigma^2 |v|^{2\delta}$ (see [52], p. 210f). To check if a generalized model that fits all species without grouping satisfies our requirements, such a model was implemented as well. So far, all models were fit using R [53] and the *nlme*-package [54]. All shortlisted models are given in Table 3, including model equation, variance model, and random effect placement.

**Table 3.** Structural representation of the first set of fitted models (urban trees only). "mm" and "gm" in the column name refer to "mixed model" and "generalized model", respectively. In the column model, the applied methodological framework is indicated, the column formula refers to the fixed effects, and column $v$ gives details about covariate and grouping applied in modeling heteroscedasticity, with $\hat{y}$ referring to fitted values (estimated above-ground biomass). Column RE indicates on which parameter random effects are finally placed. Column $\sigma$ refers to residual standard deviation.

| Name | Model | Formula | $v$ | RE | AIC | $\sigma$ |
|------|-------|---------|-----|-----|-----|-----|
| mm0 | nlme | $\alpha \cdot d1^{\beta_1}$ | d1 | a | 1982.5 | 0.057 |
| mm1 | nlme | $\alpha \cdot d1^{\beta_1} \cdot h^{\beta_2}$ | d1 $\mid$ spp | $b_1$ | 1944.8 | 0.035 |
| mm2 | nlme | $\alpha \cdot d1^{\beta_1} \cdot hgc^{\beta_3}$ | d1 $\mid$ spp | $b_1$ | 1939.2 | 0.028 |
| mm3 | nlme | $\alpha \cdot d1^{\beta_1} \cdot hgc^{\beta_3} \cdot cd^{\beta_4}$ | d1 $\mid$ spp | $b_1$ | 1907.3 | 0.030 |
| mm4 | nlme | $\alpha \cdot d1^{\beta_1} \cdot hgc^{\beta_3} \cdot cd^{\beta_4}$ | $\hat{y} \mid$ spp | $b_1$ | 1905.4 | 0.122 |
| mm5 | nlme | $\alpha \cdot d1^{\beta_1} \cdot h^{\beta_3} \cdot cd^{\beta_4}$ | d1 $\mid$ spp | $b_1$ | 1922.2 | 0.021 |
| gm6 | gnls | $\alpha \cdot d1^{\beta_1} \cdot h^{\beta_3} \cdot cd^{\beta_4}$ | d1 $\mid$ spp | - | 1940.4 | 0.012 |

For us, two questions still remain: Firstly, how do we estimate the above-ground biomass of conifer species in urban space if the fitted functions should potentially be used, e.g., in tree inventories but only include deciduous species? Secondly, is it possible to improve the predictive power of the models if additional data on forest trees are taken into account? Regarding the first question, one option is to apply biomass functions developed for traditional forests to the urban landscape, possibly incorporating an adjusting additive term or scaling factor. We tested such an approach on the available urban deciduous tree species (Table 1) using the biomass functions from the German National Forest Inventory

(NFI), which were developed for 18 species ($f_{NFI}$, c. f. [48]). We estimated $agb_{\mathrm{NFI}}$ for the available urban trees and modeled the difference as well as the ratio to the observed $agb$. Different hierarchical nonlinear models were tested, and a species-mixed allometric model (c.f. Equation (1)) also proved to be a reasonable choice, both from theoretical and practical considerations. Only $d1$ was required for modeling the absolute or relative deviations from the NFI biomass functions (notation as in Equation (1)):

$$
\begin{aligned}
y &= f_{NFI}(spp, dbh, h) + (\alpha + d1^{\beta+b}) \\
y &= f_{NFI}(spp, dbh, h) \cdot (\alpha + d1^{\beta+b})
\end{aligned}
\tag{2}
$$

In this study, both $dbh$ and $d1$ are available. If one of the two variables is not available in an application, a simple linear model can be used to predict the other one (not shown).

This leads to the second question of whether combining urban and forest data can improve the biomass functions. The idea is that a more comprehensive data basis builds a more stable allometric relation between predictors (like $d1$ and $h$) and response ($agb$) and lets the mixed-model framework statistically separate the different groups and species. Eventually, predictions for urban conifers are derivable from this statistical approach. Unfortunately, the merged data (a combination of Tables 1 and 2) miss complete information on crown attributes ($hgc$ and $cd$), as these are not available in the forest data set; therefore, only $d1$ and $h$ are the available predictors. The more challenging part in such a model is that we now need to include two factor variables (*species* and *origin*), which are not hierarchically organized (second level nested in first level) but instead are *cross classified*. This means that each species can (potentially) occur in each level of origin (Sycamore maple can be found in urban forest and traditional forests) as opposed to a hierarchical approach, where each species is constrained to a certain origin (or each origin is constrained to a certain species). We continue to use an allometric biomass model, allowing for random effects for both factors to be (potentially) included on each parameter in the model equation:

$$
y = (\alpha + a_{aoo} + a_{spp}) \cdot d1^{(\beta + b_{aoo} + b_{spp})} \cdot h^{(\gamma + c_{aoo} + c_{spp})}
\tag{3}
$$

Here again, Greek letters refer to fixed effects and the Latin letters are used for the random effect terms, which are indexed for area of origin ($aoo$) and species ($spp$). The complexity of such a nonlinear cross-classified mixed model makes convergence often difficult. Indeed, we also experienced convergence issues even with models of reduced complexity (i.e., less random effects) so that we switched from the data scale in Equation (3) to the log scale, imposing back-transformation and bias correction on predictions (e.g., "naive estimate", see [51,55]). The model equation of the final best cross-classified mixed model (CCMM), fitted using the R-package *lme4* [56], is

$$
log(y) = (\alpha + a_{aoo} + a_{spp}) + (\beta + b_{spp}) \cdot log(d1) + (\gamma + c_{spp}) \cdot h
\tag{4}
$$

using the tree height ($h$) untransformed, which yielded better results.

As a last approach, we simplified this model and included the area of origin as a binary variable, being *zero* and *one* in the case of forest and urban origin ($FU$). This reduces complexity and the 1-level mixed model could be fitted on the data scale making a bias correction obsolete but predictions for conifers in urban areas possible. The equation of this "factor" model (FM) is

$$
y = (\alpha + a_{spp} + b \cdot FU) \cdot d1^{(\beta + b_{spp})} \cdot h^{(\gamma + c_{spp})}
\tag{5}
$$

All models were examined for the urban trees using leave-one-out cross validation, both on the population and on the species level. The indicators of interest were RMSE (kg)

and BIAS (kg):

$$RMSE = \sqrt{\frac{\sum_{i=1}^{n}(y_i - \hat{y}_i)^2}{n}}$$
$$BIAS = \frac{\sum_{i=1}^{n}(y_i - \hat{y}_i)}{n}$$

(6)

Additionally, for the species-specific evaluation, we calculated relative RMSE and BIAS by dividing each by the species-specific mean *agb*. For all models, *agb* was estimated as described in the case of Equation (4); back-transformation and BIAS-correction were taken into account.

The predictions of the final CCMM model were also checked against the estimates of the NFI biomass functions [48] using the R-package *rBDAT* [45] with respect to absolute and relative RMSE and BIAS for the forest tree species of our data set. The data were used during both model developments, and hence, concerns regarding improper comparison are unfounded.

### 2.3. Sample Application

We evaluated the CCMM model using a small subset of a tree inventory containing a subset of areas in Munich, Germany, at two different points in time (2007 and 2019). All registered trees are characterized by the species name and the required model predictors. In rare cases, trees were recorded as groups, i.e., data hold the number of trees of these groups, mean diameter and height. The diameter was measured using a caliper, rounded towards full centimeter, and the height was estimated by experienced arborists and given as full meters. No measured above-ground biomass is available. Tree species, which were not covered by the set of species for which biomass functions were developed, were assigned to modeled species based on genus, habitus and expert knowledge. For few species (mainly genus *Populus*), a biomass estimation correction was applied due to very unequal specific gravity values between the model species (using *Acer*, being the best covered genus in our model data) and assigned species. For that purpose, we used the specific gravity proportion between both species according to Kollmann [43].

### 3. Results

In biomass studies, *dbh* is probably the most used predictor. But tree height is also frequently used, as it positively correlates with biomass storage (e.g., [37]). Trees of same diameter keep more biomass if larger in tree height [39,48]. When comparing urban to forest trees, urban trees show different patterns of biomass accumulation with size (see Figure 2 for the relations between *d1* and *agb*). This finding is even more pronounced if tree height is also taken into account. At the same time, tree height is very different for deciduous urban trees by a given diameter in comparison to forest trees. Based on our data, we can state that the measured deciduous urban trees are 10 to 15 m smaller than their forest counterpart (Figure 3). Interestingly, in our data, there is actually no overlap in tree height between both origins for all diameter classes. Simultaneously, applying a simple biomass model considering predictors *d1* and *h*, we find that deciduous urban trees contain more biomass for the same dimensions than forest trees. This is also shown in Figure 3 by the modeled contour lines (see also the Figure caption for an explanation and example). In consequence, we can state that deciduous urban trees usually hold less biomass than forest trees if only the diameter is considered, but if the comparison includes both diameter and height, deciduous urban trees accumulate more biomass. This seemingly contradictory result is resolved by the fact that urban trees (in our data) are usually smaller than forest trees of the same diameter class and show a different morphology. The observed difference in biomass diminishes as trees get larger and almost vanishes for trees with diameters above 100 cm (*d1*). This at least is shown by our data, although only a small share of such

large trees is present. Hence, for modeling and application, it is important to include a height measure into the biomass models.

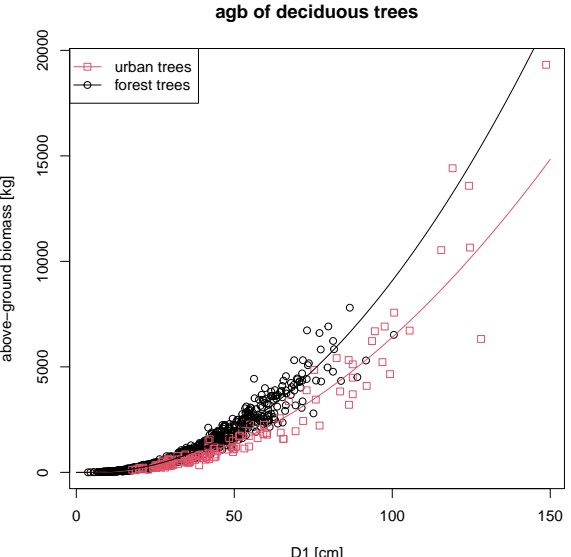

**Figure 2.** Above-ground biomass (*agb*) of deciduous forest trees (black circles) and urban trees (red squares) in relation to diameter in 1 m above ground. The continuous lines show a modeled relationship between *agb* and *d1*.

**Figure 3.** Comparison of observed (circles and squares, size proportional to above-ground biomass) and modeled *agb* (contour lines) for deciduous forest (black circles) and urban trees (red squares) with respect to *d1*, *h* and *agb*. No consideration of *species* in this graph. Each observed tree is given by its diameter and height, while *agb* (modeled by a simple allometric model using these two predictors) is given as contour lines in the background. Additionally, a functional relationship between *d1* and *h* is modeled and included (dashed lines). Horizontal and vertical lines are included to ease the interpretation of the graph: a tree with *d1* = 40 cm shows an average height of 28 m in forests and only 14 m in urban space. Of course, a forest tree given these dimensions exhibits an *agb* of approximately 1200 kg while an urban tree, which is only half in height, shows only 800 kg. Interestingly, the models indicate for trees with the comparable dimension in *d1* and *h* that urban trees theoretically exhibit more *agb* than forest trees.

We tested different approaches for fitting urban tree biomass functions. The first set of models are based on the 144 urban trees, and the second set encompasses 2205 trees,

including forest trees. We evaluated these models using AIC (first set of models [57]), model residual error, leave-one-out cross-validated root mean squared error (RMSE) and mean error (BIAS).

The first set of models, fitted using the urban tree data only, show that all four selected variables (*d1*, *h*, *hgc* and *cd*) contribute to explain the above-ground biomass of urban trees. This is coherent since all variables describe trees in their volumetric extent (see also Figure 1). The fitted mixed models had the random effect terms for factor *species* usually placed on the parameter $\beta_1$ belonging to *d1*. Heteroscedasticity (c.f. Section 2.2) was best treated also by this covariate, except for model mm4, which makes use of the model predictions to weight the errors (see Table 3). The best model by AIC (mm4, AIC = 1905.4) makes use of *d1*, *hgc* and *cd* for the fixed effects with random effects set on the exponent of *d1*. Considering the other fit statistics, i.e., residual standard error ($\sigma$) and cross-validated RMSE and BIAS (see Tables 3 and 4), other models are not much worse, or even better.

In terms of predictive performance, mm5 shows the lowest group-level cross-validated RMSE and BIAS (see Table 4, index "g" and "cv"). When it comes to applicability, the model mm1 seems to be a good choice as well because it only requires easy-to-measure quantities (*d1* and *h*, no laborious crown measurements) and exhibits only a slightly higher bias than mm5. The model results also indicate that the height of the green crown (*hgc*) might offer more information than does the tree height (*h*)—at least for model fitting. Comparing cross-validated results, it seems that using *h* instead of *hgc* delivers more accurate predictions on average (models mm1 and mm5). The model gm6, a species-independent generalized nonlinear least squares model, was fitted as a reference without an hierarchical approach and, indeed, shows the smallest residual standard error. But all other fit statistics do not show any advantage of that model.

**Table 4.** Performance for the first set of urban trees models expressed in the form of the RMSE (kg) and BIAS (kg) for the group (index *g*) as well as the population effects (index *p*) of the model fit as well as for the cross-validation (index *cv*).

| Name | $\text{rmse}_g$ | $\text{rmse}_{g,\,cv}$ | $\text{rmse}_p$ | $\text{rmse}_{p,\,cv}$ | $\text{bias}_g$ | $\text{bias}_{g,\,cv}$ | $\text{bias}_p$ | $\text{bias}_{p,\,cv}$ |
|------|------|------|------|------|------|------|------|------|
| mm0 | 695.3 | 777.9 | 804.4 | 825.3 | −36.9 | −40.7 | −80.8 | −81.6 |
| mm1 | 631.7 | 723.8 | 730.3 | 745.6 | 14.2 | 16.7 | 32.1 | 38.4 |
| mm2 | 619.9 | 714.4 | 711.2 | 738.8 | 30.1 | 27.7 | 53.3 | 55.0 |
| mm3 | 653.2 | 726.0 | 772.7 | 783.6 | 86.6 | 97.4 | 143.1 | 142.9 |
| mm4 | 664.7 | 792.4 | 764.6 | 804.5 | 91.4 | 118.5 | 135.9 | 147.7 |
| mm5 | 591.8 | 688.8 | 659.0 | 684.5 | 15.0 | 13.6 | 38.4 | 37.0 |
| gm6 | 690.8 | 740.1 | 690.8 | 740.1 | 94.6 | 100.0 | 94.6 | 100.0 |

As mentioned above, our full data clearly indicate that there is a difference in the above-ground biomass between urban and forest trees of the same diameter class. Modeling the deviation between urban and forest biomass to be able to use well-approved forest biomass functions using an additive or multiplicative correction factor does not lead to satisfactory results: this type of model, based on adjusting forest tree biomass functions, turns out to have the highest $\text{RMSE}_{cv}$ values of all tested models and moderately high $\text{BIAS}_{cv}$ values (see Table 5). The additive model shows lower $\text{RMSE}_{cv}$ but higher $\text{BIAS}_{cv}$ than the multiplicative model.

The CCMM model of Equation (4), an extension of the well-suited mm1 model (with predictors *d1* and *h*) with both urban and forest trees, shows best values for both $\text{RMSE}_{cv}$ and $\text{BIAS}_{cv}$. Lastly, the simplified FM model, implementing Equation (5) using a binary variable encoding forest and urban origin, exhibits a $\text{BIAS}_{cv}$ high as −90.6 kg (almost nine times higher than the best model in absolute values) and moderately high $\text{RMSE}_{cv}$ (see Table 5).

Summing up, the best-performing model for urban trees is the CCMM. To assure that the CCMM also reproduces observed biomass of forest trees equally well, we compared the bias-adjusted predictions with the predictions of the German NFI functions [48] for Norway spruce, Scots pine, Douglas fir, European beech, oak and sycamore. Results indicate slightly

higher RMSE values for all species except Douglas fir and sycamore ($-26.9\%$ to $+17.8\%$) and smaller absolute BIAS values for all species, except for Norway spruce (NFI: $+8.0$ kg, CCMM: $-12.8$ kg). Highest (in absolute terms) relative BIAS of the CCMM model is $-3.2\%$.

When evaluating the different models for particular species, results might show different patterns (see Tables A1 and A2 in Appendix A for absolute and relative cross-validated RMSE and BIAS). In this case, no model exhibits best results for all of the 14 considered deciduous tree species. The "best" models show smallest $BIAS_{cv}$ values for only four species and smallest $RMSE_{cv}$ values for only five species, respectively. Still, the CCMM shows very good results: it is the only model with all relative BIAS values below 10% and relative RMSE ranges between 10 and 36% (cf. Table A2). Similarly, mm1 and mm5 models also perform well, but the largest BIAS is higher (for sycamore and birch, respectively). Only for black locust is the result of mm1 better than that of CCMM considering RMSE and BIAS. But both models (mm1 and mm5) exceed 10% BIAS for 3 and 4 tree species, respectively. Moreover, both models can only predict deciduous trees. This is an advantage of the CCMM model because coniferous trees in urban areas can be represented by the integration of coniferous tree species data and the methodology of cross-classified mixed models. An independent evaluation of the performance for these additional tree species in the city is not possible with our data. But it has been shown that the model performs well in all testable aspects (deciduous urban trees as well as coniferous and deciduous forest trees).

In summary, the finally proposed CCMM model (Equation (4)) uses only *d1* and *h* as predictors due to data limitations, but includes independent random effects given *species* for the intercept and the two parameters. Random effects based on the *origin* (traditional forest vs. urban area) were significant only for the intercept ($a_{aoo}$), making it a scaling parameter (this finding lead to the development of the factor model of Equation (5)). The estimated parameters for the CCMM model are given in Table 6, and the random effects for different species are given in Table A3 in Appendix B. The factor for correcting bias is estimated to be 1.012081 (see [51,55]). As long as predictions correspond to the group level, i.e., refer to a particular species and location as in our example, the given approach for bias correction by $e^{0.5\sigma^2}$ is valid and corrects for the bias of transformation of the common within-group error. For prediction on population level (i.e., setting the random effects to zero), the uncertainty of the between-group error must be incorporated as well (e.g., see Appendix 1 of [58]).

**Table 5.** Comparison of the different modeling approaches by RMSE (kg) and BIAS (kg). "mm1" and "mm5" refer to the mixed models from Table 3 without forest trees and with, respectively, without crown diameter (*cd*). "FB+D" and "FB*R" are adjusted forest tree biomass functions (Equation (2)), "CCMM" refers to the cross-classified mixed model (Equation (4)) and "FM" is the factor model from Equation (5). Column headings "n" show the number of observations of model fit (index *fit*) and for the cross-validation (index *cv*), respectively. The indices "dec" und "con" refer to deciduous and conifer tree species. Cross-validated RMSE and BIAS shown here refer to the group level of the models.

| No | Model | $n_{fit}$ | $n_{cv}$ | $n_{dec}$ | $n_{con}$ | RMSE | Bias | $RMSE_{cv}$ | $Bias_{cv}$ |
|----|-------|-----------|----------|-----------|-----------|--------|--------|-------------|-------------|
| 1 | mm1 | 144 | 144 | 14 | 0 | 631.7 | 14.2 | 723.8 | 16.7 |
| 2 | mm5 | 144 | 144 | 14 | 0 | 591.8 | 15.0 | 688.8 | 13.6 |
| 3 | FB + D | 144 | 144 | 14 | 0 | 790.1 | 22.0 | 802.2 | 21.7 |
| 4 | FB*R | 144 | 144 | 14 | 0 | 842.1 | 32.7 | 849.4 | 12.8 |
| 5 | CCMM | 2205 | 144 | 15 | 4 | 598.6 | $-10.7$ | 667.1 | $-10.7$ |
| 6 | FM | 2205 | 144 | 15 | 4 | 693.1 | $-76.8$ | 780.9 | $-90.6$ |

**Table 6.** Parameter estimates of the CCMM model from Equation (4). The first three columns ($\alpha$, $\beta$ and $\gamma$) give the estimated fixed effects. Column four to seven give the estimated standard deviations of the independent random effect terms. $\sigma$ gives the residual standard deviation of the model. Estimated random effects for different species can be found in the Table A3 in Appendix B.

| Model | $\alpha$ | $\beta$ | $\gamma$ | $Std.a_{spp}$ | $Std.b_{spp}$ | $Std.c_{spp}$ | $Std.a_{aao}$ | $\sigma$ |
|---|---|---|---|---|---|---|---|---|
| CCMM | −1.50880 | 2.02329 | 0.03487 | 0.13843 | 0.04568 | 0.00349 | 0.16024 | 0.15498 |

*Sample Application*

To show the applicability of the developed model, we applied the CCMM with iSiMan5 tree management software [59] to a small subset of data collected during a regular inventory by TreeConsult Brudi & Partner in a residential area in the city of Munich, Germany. The model estimates the stored above-ground biomass *agb* of the urban trees, which can also be easily translated into stored amount of carbon by applying the respective carbon content factor (see, e.g., [60,61]). If data for multiple points in time are available, in this case for 2007 and 2019 (see Table 7), it even is possible to calculate the respective net carbon fluxes.

**Table 7.** Carbon storage and *agb* for the example area in Munich by urban trees for 2007 and 2019. Indices "m" and "med" refer to mean and median values. A clear increase in storage can be highlighted (+10.2%) despite a decrease in the number of trees.

| Year | Stems | agb | C | $CO_2$ | $d1_m$ | $h_m$ | $agb_m$ | $d1_{med}$ | $h_{med}$ |
|---|---|---|---|---|---|---|---|---|---|
| [a] | [n] | [kg] | [kg] | [kg] | [cm] | [m] | [kg] | [cm] | [m] |
| 2007 | 2763 | 854,176 | 411,950 | 1,511,856 | 22.6 | 10.2 | 309.1 | 19.1 | 9.0 |
| 2019 | 2260 | 941,050 | 453,472 | 1,664,241 | 26.4 | 10.2 | 416.4 | 23.9 | 9.0 |

The results for our example data show that in 2019, the urban trees stored about 941 tons of biomass, i.e., about 453 tons of carbon (c.f. Figure 4). This amount of storage was achieved by 2260 trees, meaning an average amount of stored biomass of 416 kg per tree. Compared to 2007, this is an increase of about 86 tons of biomass or a plus of 10.2%. The average sink capacity for this set of trees is thus approximately 3.5 tons of carbon per year, even though 503 trees were lost at the same time (−18.2%). By an increased average diameter and constant tree height, the loss in numbers could be overcompensated in terms of carbon storage. This highlights again the importance of managing, tending and conserving especially middle-aged and old urban trees, which not only act as a carbon storage but also serve further ecosystem services (social, economic, ecological, climatic and aesthetical, see further, e.g., [1,3]). Since the developed equation of CCMM takes into account the different tree species and their sizes, further analyses in combination with the data from the tree inventory are possible.

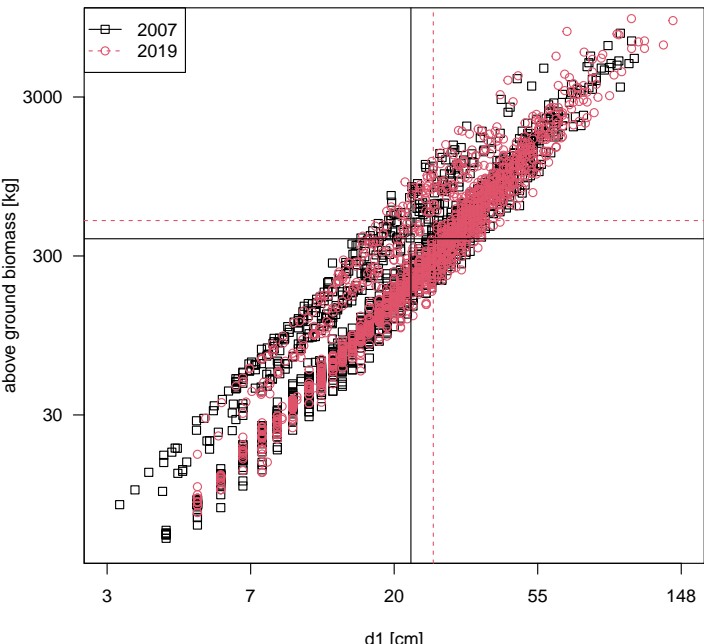

**Figure 4.** Predicted above ground biomass of the Munich data against *d1* on double log scale. Black squares indicate 2007 data and red circles refer to 2019 data. Vertical and horizontal lines mark the average biomass and diameter. The two-part pattern is due to differences in tree heights and applies to almost all tree species. On average, tree height remains constant, and biomass increase is driven by diameter increase.

## 4. Discussion

There are several other, supposedly more modern, approaches to determine the biomass and C-sink potential of urban trees (e.g., aerial and terrestrial laser scanning). But the use of the close allometric relationships between simple-to-measure tree attributes and the target variables biomass and carbon storage, especially in combination with regular and repeated inventory data, remains an accurate and low-cost method to determine urban tree carbon storage. In addition, the method enables more modern techniques (e.g., terrestrial laser scanning) to expand the data basis for model building.

Our data analysis shows that a simple transfer of forest biomass functions into urban space needs adjustment because the allometry of urban trees differs strongly from those of forest trees. Although the same tree size in terms of diameter (e.g., *d1*) can be found in both landscapes, tree heights (and also crown habitus) differ significantly. In our data set, there was actually no overlap in tree heights for any given diameter class, so the two origins should be considered separate entities. Still, both are made up by trees, obeying allometric rules. Differences in biomass between urban and forest trees are more prominent for smaller tree dimensions and diminish as trees mature. This is probably because urban trees grow under less light competition in less confined spaces and therefore grow less tall and form larger crowns right from the very beginning. The larger trees get, both origins resemble a more unconstrained habitus and thus show more comparable biomass and carbon storage. As a clear result, we can state that there is a need for urban tree biomass functions in addition to forest tree biomass functions.

The application of forest biomass models in an urban setting is an extrapolation of those models, which requires a significant correction. Our approaches using the NFI biomass functions, including correction, performed worse in comparison to most other tested models. The models, developed for urban tree biomass, make use of the predictors *d1* and *h*, which are regularly measured during urban tree inventories. In the case of using a diameter measured in a different height, e.g., in 1.3 m (*dbh*) as regularly used in forestry, a simple linear regression can convert between both variables due to high correlation (see,

e.g., [62]). It turns out by model cross validation that additional variables like height of green crown (hgc) or crown diameter (cd)—although significant during model fitting—do not or only slightly improve the models. Actually, our CCMM model shows best overall statistics only requiring *d1* and *h*.

We propose to use the CCMM model, which fits several species and origins (urban and forest areas) at once. This model shows the best performance compared to models only using data from urban areas. Hence, this model "learns" from forest tree data and improves the modeled relationship. Additionally, it is possible to estimate yet unmeasured crossings (e.g., conifer species in urban areas). Due to the lack of data, this case cannot be validated, but we could show that the model behaves well in all statistically verifiable aspects. It predicts forest biomass equally well as the NFI functions and outperforms the urban tree-only models by better capturing the general allometric relation. But of course, the difference between urban trees and forest trees is contained only in the random effect of the scaling parameter. Thus, a similar difference between these two groups of tree species of the two origins is assumed since both deciduous and coniferous trees have more space and less competition in urban areas than in the traditional forest environment. Hence, crown formation and height growth are likely to be modified in a similar direction for both groups.

The CCMM uses the mixed-effects modeling framework. From a theoretical perspective, the validity of a model is assured, among other things, by checking the assumption of normality for the estimated random effects. This is difficult if the factor variable has only two levels, and often 5–6 factor levels are recommended as a minimum. Nevertheless, the case of less factor levels is not uncommon (e.g., female vs. male), and some authors see no reason to prevent the use of mixed models in such cases (e.g., [63], p. 247/275f). Our results indicate the convergence and suitable parameter estimates of the CCMM model as well as proper fit statistics so that we can assume the correctness and applicability of the model.

The set of (urban) tree species included reflects the situation in Karlsruhe, Germany in 2011, and hence, it is not necessarily representative for other cities, not even in Germany. Some tree species are represented only by a small number of samples, so it is important to enhance our data by more tree species, especially if future development may be directed toward more suitable tree species in a changing urban climate. There might also be differences within tree species depending on different urban regions (c.f. [34]), requiring more specific and localized models. Using the mixed-models approach, new tree species (or subspecies with special habitus like *Populus nigra italica*) can be added into the model by estimating their random effect even with a few observations only so that local situations can be handled with little effort [64]. Besides that, collecting data from different cities—here, TLS might be a suitable technique—might help with building a more general model, which includes random effect terms for different cities and makes it possible to expand the model geographically. The data of the forest trees instead originate from different studies (both with regard to the geographic origin and sampling method).

As an example, we applied the CCMM model to a subset of a tree inventory in Munich, Germany. The assignment of the encountered tree species to the model tree species, as well as the regional transfer of the model, eventually leads to a certain inaccuracy of the result. Yet, this is unavoidable, as the corresponding basic data are missing so far.

With the application of the model, and despite its certain inaccuracies in application, it was possible to represent carbon storage at two different points in time and, thus, estimate carbon fluxes over the period under consideration. The closer look at the performance of the stand or even of the individual species or genera, also on different site types, can thereby provide important advice for future tree management in urban areas, taking climate protection and climate adaptation into account.

## 5. Conclusions

Biomass functions are an easy and low-cost method to estimate biomass and carbon storage, both in forests and urban landscapes. Although trees are the main carbon sink and obey close allometric relationships in both ecosystems, the simple use of widely available forest biomass functions in urban areas is not recommended. Our data show clear differences in dimensions and biomass allocation. Hence, the development of specific urban biomass functions is important. Based on data from urban and forest areas, we present a new biomass model fitted for multiple species and origins at once. This model is capable of differentiating between both origins and at the same time improving predictive power compared to single-origin-models, i.e., the model "learns" from the allometric relation of forest trees. The cross-classified mixed model (CCMM) can also estimate unobserved groups, which in this case are conifers in urban areas. We checked the model in all verifiable aspects and concluded that the model performs well both for urban as well as forest trees. The assumption of comparable differences in habitus and biomass allocation between urban and forest trees for both deciduous and conifer species needs to be further investigated. The presented approach deserves further attention because it can easily be extended, e.g., by including a city-group level or random effects estimation for further tree species by means of the mixed-effects modeling framework. As an exemplary application, we use the preferred CCMM model to estimate carbon storage in the area of some residential neighborhoods in Munich for two points in time, showing an increase in biomass and carbon storage despite a reduced number of stems and constant average tree height. The main driver in this example is an average increase of diameter of about 4 cm within 12 years. With such data on carbon storage and fluxes, management options can be developed and evaluated. Further work should concentrate on currently less frequent species, especially in view of rapid climate change and subsequent changes in tree species composition. With that, a closer look into the carbon sink potential of different species over time would be possible and improve knowledge of ecosystem services provided by urban trees.

**Author Contributions:** Conceptualization, C.V. and A.A.; methodology, formal analysis, and validation, C.V.; data curation, A.A. and C.V.; writing—original draft preparation, C.V.; writing—review and editing, A.A. and C.V.; visualization, C.V. All authors have read and agreed to the published version of the manuscript.

**Funding:** This research received no external funding.

**Data Availability Statement:** The data presented in this study are available on request from the corresponding author.

**Acknowledgments:** Thanks are due to three anonymous reviewers whose comments contributed significantly to a better understanding and integration of this work and its results. Special thanks go also to colleagues helping improve the structure and comprehensibility of the manuscript.

**Conflicts of Interest:** The authors declare no conflict of interest.

## Appendix A. Cross-Validated Species-Specific Results

**Table A1.** Overview on cross-validated measures of the different model approaches. The abbreviations "R" and "B" refer to RMSE (kg) and BIAS (kg), respectively. The corresponding indices relate to column "no" of Table 5.

| Species | n | $R_1$ | $B_1$ | $R_2$ | $B_2$ | $R_3$ | $B_3$ | $R_4$ | $B_4$ | $R_5$ | $B_5$ | $R_6$ | $B_6$ |
|---|---|---|---|---|---|---|---|---|---|---|---|---|---|
| sycamore maple | 3 | 191.6 | −172.9 | 216.7 | −184.6 | 167.2 | 25.9 | 166.1 | 4.0 | 262.6 | 88.0 | 263.7 | 60.1 |
| birch spp. | 3 | 251.3 | −133.5 | 210.4 | −190.0 | 228.7 | −96.3 | 218.8 | −98.0 | 258.2 | −5.3 | 137.1 | −137.1 |
| oak spp. | 21 | 381.2 | −49.3 | 562.6 | −99.9 | 243.5 | 85.0 | 321.8 | 97.4 | 623.9 | −149.9 | 963.0 | −286.2 |
| common ash | 15 | 549.4 | −14.2 | 675.8 | 9.7 | 274.9 | 4.9 | 360.8 | −61.0 | 497.4 | −47.0 | 753.7 | −193.8 |
| field maple | 5 | 40.5 | 5.6 | 43.2 | 4.7 | 32.1 | 6.3 | 35.4 | 14.5 | 40.6 | −2.1 | 44.4 | 1.1 |
| common hornbeam | 12 | 137.0 | −16.4 | 139.0 | −12.9 | 152.1 | −22.3 | 165.0 | −33.2 | 164.1 | −9.4 | 259.5 | −59.5 |
| Caucasian lime | 7 | 449.2 | −181.9 | 241.7 | −98.4 | 708.5 | −234.7 | 890.6 | −318.3 | 386.2 | −105.9 | 576.8 | −235.0 |
| London plane | 14 | 1909.5 | 182.8 | 1523.3 | −67.5 | 2269.8 | 109.2 | 2381.7 | 168.9 | 1633.9 | 98.4 | 1669.5 | −11.2 |
| black locust | 6 | 344.5 | −86.6 | 252.6 | 144.9 | 101.6 | 75.1 | 81.9 | 52.1 | 426.4 | −136.1 | 733.0 | −243.4 |
| horse chestnut | 6 | 450.5 | 216.2 | 497.5 | 251.8 | 356.2 | −89.2 | 402.1 | 24.8 | 432.0 | 150.3 | 394.8 | 152.9 |
| red oak | 8 | 1207.6 | 402.9 | 1363.2 | 449.6 | 1166.3 | 274.1 | 1167.3 | 208.0 | 898.6 | 236.9 | 839.2 | 116.6 |
| Norway maple | 32 | 346.4 | −18.8 | 342.6 | 11.3 | 270.0 | 0.7 | 250.3 | −8.4 | 368.6 | −13.2 | 401.4 | −53.3 |
| wild cherry | 4 | 78.1 | −8.2 | 98.6 | 1.5 | 49.2 | −9.9 | 56.4 | −17.0 | 60.0 | −3.8 | 76.6 | −0.7 |
| small−leafed lime | 8 | 70.6 | 8.1 | 60.9 | 6.2 | 99.5 | −32.0 | 196.3 | −74.9 | 40.9 | 1.5 | 33.0 | 3.4 |

**Table A2.** Overview on cross-validated measures of the different model approaches. The abbreviations "rR" and "rB" refer to the relative RMSE (coefficient of variation) and relative BIAS. The corresponding indices relate to column "no" of Table 5. Relative RMSE and BIAS are scaled by species-specific mean observed above-ground biomass.

| Species | n | $rR_1$ | $rB_1$ | $rR_2$ | $rB_2$ | $rR_3$ | $rB_3$ | $rR_4$ | $rB_4$ | $rR_5$ | $rB_5$ | $rR_6$ | $rB_6$ |
|---|---|---|---|---|---|---|---|---|---|---|---|---|---|
| sycamore maple | 3 | 15.1 | −13.6 | 17.1 | −14.6 | 13.2 | 2.0 | 13.1 | 0.3 | 20.7 | 6.9 | 20.8 | 4.7 |
| birch spp. | 3 | 29.0 | −15.4 | 24.3 | −21.9 | 26.4 | −11.1 | 25.2 | −11.3 | 29.8 | −0.6 | 15.8 | −15.8 |
| oak spp. | 21 | 21.8 | −2.8 | 32.1 | −5.7 | 13.9 | 4.9 | 18.4 | 5.6 | 35.6 | −8.6 | 55.0 | −16.3 |
| common ash | 15 | 26.0 | −0.7 | 32.0 | 0.5 | 13.0 | 0.2 | 17.1 | −2.9 | 23.6 | −2.2 | 35.7 | −9.2 |
| field maple | 5 | 19.6 | 2.7 | 20.9 | 2.3 | 15.6 | 3.0 | 17.2 | 7.0 | 19.7 | −1.0 | 21.5 | 0.6 |
| common hornbeam | 12 | 24.0 | −2.9 | 24.4 | −2.3 | 26.7 | −3.9 | 28.9 | −5.8 | 28.8 | −1.7 | 45.5 | −10.4 |
| Caucasian lime | 7 | 31.8 | −12.9 | 17.1 | −7.0 | 50.2 | −16.6 | 63.0 | −22.5 | 27.3 | −7.5 | 40.8 | −16.6 |
| London plane | 14 | 41.5 | 4.0 | 33.1 | −1.5 | 49.3 | 2.4 | 51.8 | 3.7 | 35.5 | 2.1 | 36.3 | −0.2 |
| black locust | 6 | 24.5 | −6.2 | 18.0 | 10.3 | 7.2 | 5.3 | 5.8 | 3.7 | 30.3 | −9.7 | 52.1 | −17.3 |
| horse chestnut | 6 | 27.2 | 13.0 | 30.0 | 15.2 | 21.5 | −5.4 | 24.2 | 1.5 | 26.0 | 9.1 | 23.8 | 9.2 |
| red oak | 8 | 22.7 | 7.6 | 25.6 | 8.5 | 21.9 | 5.2 | 21.9 | 3.9 | 16.9 | 4.5 | 15.8 | 2.2 |
| Norway maple | 32 | 25.2 | −1.4 | 25.0 | 0.8 | 19.7 | 0.1 | 18.2 | −0.6 | 26.9 | −1.0 | 29.2 | −3.9 |
| wild cherry | 4 | 25.3 | −2.7 | 32.0 | 0.5 | 15.9 | −3.2 | 18.3 | −5.5 | 19.4 | −1.2 | 24.8 | −0.2 |
| small−leafed lime | 8 | 18.0 | 2.1 | 15.5 | 1.6 | 25.4 | −8.2 | 50.1 | −19.1 | 10.4 | 0.4 | 8.4 | 0.9 |

## Appendix B. Parameter Estimates of the CCMM

**Table A3.** Parameter estimates of the cross-classified mixed model of Equation (4) for different species. Parameter $a_{aoo}$ would take the value −0.15417 in case of forest trees.

| Species | $\alpha$ | $a_{aao}$ | $a_{spp}$ | $\beta$ | $b_{spp}$ | $\gamma$ | $c_{spp}$ |
|---|---|---|---|---|---|---|---|
| birch spp. | −1.50880 | 0.15417 | −0.05195 | 2.02329 | −0.01990 | 0.03487 | −0.00017 |
| black locust | −1.50880 | 0.15417 | 0.13679 | 2.02329 | 0.04328 | 0.03487 | 0.00066 |
| Caucasian lime | −1.50880 | 0.15417 | −0.02768 | 2.02329 | −0.02594 | 0.03487 | 0.00012 |
| common ash | −1.50880 | 0.15417 | 0.12836 | 2.02329 | 0.01542 | 0.03487 | 0.00143 |
| common hornbeam | −1.50880 | 0.15417 | 0.01220 | 2.02329 | 0.01543 | 0.03487 | 0.00200 |
| douglas fir | −1.50880 | 0.15417 | −0.09424 | 2.02329 | −0.01337 | 0.03487 | −0.00441 |
| European beech | −1.50880 | 0.15417 | −0.01902 | 2.02329 | 0.05054 | 0.03487 | 0.00331 |
| field maple | −1.50880 | 0.15417 | 0.04772 | 2.02329 | 0.01217 | 0.03487 | 0.00015 |
| horse chestnut | −1.50880 | 0.15417 | 0.00105 | 2.02329 | 0.02287 | 0.03487 | 0.00038 |
| London plane | −1.50880 | 0.15417 | −0.01711 | 2.02329 | −0.03266 | 0.03487 | 0.00054 |
| Norway maple | −1.50880 | 0.15417 | 0.02568 | 2.02329 | 0.00816 | 0.03487 | 0.00147 |
| Norway spruce | −1.50880 | 0.15417 | 0.18834 | 2.02329 | −0.10420 | 0.03487 | −0.00453 |
| oak spp. | −1.50880 | 0.15417 | 0.01319 | 2.02329 | 0.04164 | 0.03487 | 0.00117 |
| red oak | −1.50880 | 0.15417 | 0.04590 | 2.02329 | 0.02222 | 0.03487 | 0.00134 |
| Scots pine | −1.50880 | 0.15417 | −0.28100 | 2.02329 | −0.00441 | 0.03487 | 0.00027 |
| silver fir | −1.50880 | 0.15417 | −0.03706 | 2.02329 | −0.02470 | 0.03487 | −0.00202 |
| small−leafed lime | −1.50880 | 0.15417 | −0.10021 | 2.02329 | −0.03197 | 0.03487 | −0.00072 |
| sycamore maple | −1.50880 | 0.15417 | −0.00223 | 2.02329 | 0.01602 | 0.03487 | −0.00079 |
| wild cherry | −1.50880 | 0.15417 | 0.03127 | 2.02329 | 0.00939 | 0.03487 | −0.00017 |

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
