# Peer review of "Learning from Forest Trees: Improving Urban Tree Biomass Functions"

_forests, doi:10.3390/f14071473_

Round 1
Reviewer 1 Report
Dear authors, it's very well researched. What is missing is the information regarding the growth area of urban trees. It refers to tree rows or trees with a large growing area in parks. This information is not given in the survey carried out. This information is vital because it affects the growth of trees in the urban area.
Also, the season in which the data was taken is not mentioned.
Text editing lines 94, 121, 159, 209,
Author Response
Review 1:
Comments and Suggestions for Authors
Dear authors, it's very well researched. What is missing is the information regarding the growth area of urban trees. It refers to tree rows or trees with a large growing area in parks. This information is not given in the survey carried out. This information is vital because it affects the growth of trees in the urban area.
Also, the season in which the data was taken is not mentioned.
Response: Thank you very much for this valuable comment. We added information about the trees, the selection process and the sampling time.
Text editing lines 94, 121, 159, 209,
Response: We edited the marked lines by more closely defining the sentences.
Reviewer 2 Report
Line 48. several remote sensing make full use of...what do you mean by it. Please correct and refine the sentence
Line 106, please avoid starting sentence with "because'.
Line 143 and Line 146, please use Figure...and not figure
For your results, all your tables and figures were using the comparison between urban trees and forest trees. It is confusing, as urban trees also comes from an urban forest, hence, a forest as well. Unless those urban trees are the one in the middle of the city. Please give justification on this matter, or if not, please change the name of those 2 comparisons. For example, 'urban forest trees' and ' virgin forest trees'
Minor editing of English is required, as stated and highlighted in the reviews
Author Response
Comments and Suggestions for Authors
Line 48. several remote sensing make full use of...what do you mean by it. Please correct and refine the sentence
Response: We re-phrased the sentence and improved clarity and causality.
Line 106, please avoid starting sentence with "because'.
Response: rephrased
Line 143 and Line 146, please use Figure...and not figure
Response: Updated figure to Figure and table to Table. (Not highlighted in the edited text by red color).
For your results, all your tables and figures were using the comparison between urban trees and forest trees. It is confusing, as urban trees also comes from an urban forest, hence, a forest as well. Unless those urban trees are the one in the middle of the city. Please give justification on this matter, or if not, please change the name of those 2 comparisons. For example, 'urban forest trees' and ' virgin forest trees'
Response: Thank you very much for your comments, especially about the necessity to closer define “urban trees” and “forest trees”. We added a paragraph to specify both terms in the context of our study.
Comments on the Quality of English Language
Minor editing of English is required, as stated and highlighted in the reviews
Response: we edited the stated and highlighted paragraphs.
Reviewer 3 Report
1. The format of the author information is incorrect.
2. The abstract should discuss the Purpose, Methods, Results, and Conclusion respectively. It is recommended that the author further organize and condense the content and expression in the abstract section.
3. Lines 44-54: The applicability and advantages/disadvantages of these different methods should be briefly introduced and explained in relation to this study.
4. Lines 55-59: I believe this paragraph can be included in lines 63-73 to further explain the research methodology of this experiment.
5. Lines 60-62: It is suggested to enrich the description of the biomass function in this section, such as providing background information and explaining the reasons for choosing the biomass function as the research methodology in this study.
6. All tables are missing units.
7. Lines 93-94: "graphical overview" can be placed in the Results section.
8. Lines 105-115: There are errors in the line numbering, and it is recommended that the author carefully review the formatting.
9. Allometric models should be briefly introduced.
10. In Methods, how did the author determine the inclusion of two factor variables (species and origin) that are not hierarchically organized but cross-classified? This point needs further explanation.
11. In the Methods section, it is recommended to organize the author's thought process into steps for determining the model.
12. Line 111: There is no information in the table related to d1 or agb. Perhaps further explanation is needed here to express what the author intends to convey.
13. In Methods, when mentioning "how to treat conifer species in urban space if the functions should potentially be used," what does "the functions" refer to? The author did not explain it clearly.
14. The mentioned part about equation 1 does not include the biomass function and d1 related to NFI. Perhaps it refers to equation 2? Additionally, equation 3 mentioned later in the text is not referenced anywhere in the paper. It is advised that the author carefully check the correct citation and explanation of the equations in the text.
15. Line 138: Figure 2 reflects the relationship between d1 and agb, not dbh.
16. Line 149: What does "thickness" refer to? It is not explained in the text.
17. Lines 138-156: The importance of incorporating height into the biomass model has been demonstrated, but it is only compared between forest trees and urban trees at different heights, without explaining the impact of different heights on biomass production for trees of the same origin.
The overall quality of English language usage in the manuscript is good. The sentences are well-structured, and the ideas are conveyed effectively. However, there are a few areas where minor improvements can be made to enhance clarity and readability that will enhance the clarity and flow of the text while maintaining the intended meaning.
Author Response
Comments and Suggestions for Authors
- The format of the author information is incorrect.
Response: We followed the instruction of the Latex-template and if required, we will stay in touch with the editors to correct the format.
- The abstract should discuss the Purpose, Methods, Results, and Conclusion respectively. It is recommended that the author further organize and condense the content and expression in the abstract section.
Response: We restructured the abstract accordingly.
- Lines 44-54: The applicability and advantages/disadvantages of these different methods should be briefly introduced and explained in relation to this study.
Response: We updated this paragraph.
- Lines 55-59: I believe this paragraph can be included in lines 63-73 to further explain the research methodology of this experiment.
Response: we edited lines 55 – 73 jointly so that point 4, 5 and 9 should be better contextualised in total.
- Lines 60-62: It is suggested to enrich the description of the biomass function in this section, such as providing background information and explaining the reasons for choosing the biomass function as the research methodology in this study.
Response: see point 4
- All tables are missing units.
Response: we added units for table 1 + 2 and added information of the units of RMSE and BIAS.
- Lines 93-94: "graphical overview" can be placed in the Results section.
Response: We consider the graphical overview (Fig. 1) to stand in parallel to Table 1 and to be part of the data presentation, not of the results.
- Lines 105-115: There are errors in the line numbering, and it is recommended that the author carefully review the formatting.
Response: I assume, this is a bug in the latex class and cannot be resolved by us. It is automatically generated when running the .tex file. Should by no problem for the final version or be resolved by the editors/typesetters.
- Allometric models should be briefly introduced.
Response: See point 5
- In Methods, how did the author determine the inclusion of two factor variables (species and origin) that are not hierarchically organized but cross-classified? This point needs further explanation.
Response: We added some clarification to the decision, why the two factors are not nested but crossed.
- In the Methods section, it is recommended to organize the author's thought process into steps for determining the model.
Response: We added a paragraph introducing the modelling steps in short.
- Line 111: There is no information in the table related to d1 or agb. Perhaps further explanation is needed here to express what the author intends to convey.
Response: We added more information to the citation of table 3 to make clear, which variance covariate was used.
- In Methods, when mentioning "how to treat conifer species in urban space if the functions should potentially be used," what does "the functions" refer to? The author did not explain it clearly.
Response: We clarified the raised question.
- The mentioned part about equation 1 does not include the biomass function and d1 related to NFI. Perhaps it refers to equation 2? Additionally, equation 3 mentioned later in the text is not referenced anywhere in the paper. It is advised that the author carefully check the correct citation and explanation of the equations in the text.
Response: We improved to description of eq.1, and added reference to Table 3, where all shortlisted models of the general from of eq.1 are shown. Indeed, eq3 is not mentioned in the text anywhere, because it is used as a starting point and to show the allometric nature of the formula. We have updated the introduction to this formula.
- Line 138: Figure 2 reflects the relationship between d1 and agb, not dbh.
Response: corrected.
- Line 149: What does "thickness" refer to? It is not explained in the text.
Response: “thickness” replaced by “diameter”
- Lines 138-156: The importance of incorporating height into the biomass model has been demonstrated, but it is only compared between forest trees and urban trees at different heights, without explaining the impact of different heights on biomass production for trees of the same origin.
Response: We added some explanation on the effect of tree height.
Comments on the Quality of English Language
The overall quality of English language usage in the manuscript is good. The sentences are well-structured, and the ideas are conveyed effectively. However, there are a few areas where minor improvements can be made to enhance clarity and readability that will enhance the clarity and flow of the text while maintaining the intended meaning.
Response: Thank you. We hope that the editing of the manuscript improved readability and comprehensibility of our study.
Round 2
Reviewer 3 Report
The authors have responded to all the questions I raised